# A Non-Flat Terrain Biped Gait Planner Based on DIRCON

**DOI:** 10.3390/biomimetics7040203

**Published:** 2022-11-18

**Authors:** Boyang Chen, Xizhe Zang, Yue Zhang, Liang Gao, Yanhe Zhu, Jie Zhao

**Affiliations:** State Key Laboratory of Robotics and System, Harbin Institute of Technology, Harbin 150001, China

**Keywords:** robotics, biped gait planning, trajectory optimization, non-flat terrain

## Abstract

Various constraints exist in bipedal movement. Due to the natural ability of effectively handling constraints, trajectory optimization has become one of the mainstream methods in biped gait planning, especially when constraints become much more complex on non-flat terrain. In this paper, we propose a multi-modal biped gait planner based on DIRCON, which can generate different gaits for multiple, non-flat terrains. Firstly, a virtual knot is designed to model the state transitions when the swing foot contacts terrain and is inserted as the first knot of the target trajectory of the current support phase. Thus, a complete gait or multi-modal gaits sequence can be generated at one time. Then, slacked complementary constraints, which can avoid undesired trajectories, are elaborated to describe the coupling relationships between terrain information and bipedal motion for trajectory optimization based gait planning. The concrete form of the gait planner is also delivered. Finally, we verify the performance of the planner, as well as the structural design of our newly designed biped robot in CoppeliaSim through flat terrain walking, stairs terrain walking and quincuncial piles walking. The three experiments show that the gaits planned by the proposed planner can enable the robot to walk stably over non-flat terrains, even through simple PD control.

## 1. Introduction

Robots are subjected to various constraints during the movement process, such as joint limits, obstacle avoidance and external contact constraints. Because the base of a biped robot floats while the supporting foot must maintain contact with the terrain and not slide, more constraints should be considered in the biped gait planning. Especially when the terrain is non-flat, such as stairs or quincuncial piles, the constraints become more stringent.

Currently, Hybrid Zero Dynamics (HZD) [1,2,3], Reinforce Learning (RL) [4,5,6,7] and Trajectory Optimization (TO) [8,9,10,11,12,13,14,15,16] are the three mainstream methods for the biped gait planning. Because of the natural ability to deal with multifarious constraints, the TO method is particularly suitable for much more complex constraints in biped gait planning [10,12]. At the moment, flat terrain gait planning based on TO has produced a number of promising experimental results. Michael Posa et al. [10] used Direct Transcription (DT) to generate a joint-level trajectory that allows the FastRunner to run at 20 MPH without specifying the contact sequence in advance. Hereid et al. [13] introduced an on-line gait planner based on Direct Collocation (DC) without removing or lumping degrees of freedom, which makes the Cassie series robot walk dynamically.

However, as the non-flat terrain further limits robot movement and the requirement of balance maintenance becomes much more difficult, more motion details and constraints need to be considered during biped gait planning to acquire a dynamic, feasible trajectory. Up to now, biped gait planning of non-flat terrain is still an open problem. Li et al. [17] proposed a Linear Inverted Pendulum Model (LIPM)-based non-flat terrain gait planner with the consideration of both the single support phase and double support phase, and the robot can step over a convex terrain. Although the reduced dynamic model, such as the LIPM, can make the robot adapt to non-flat terrain, it is difficult to apply to the large-scale structural non-flat terrain like a staircase or quincuncial piles because of the requirement of the constant height of center of mass, and only a few motion details can be considered. Kumar et al. [18] trained a feedforward neural network with a gait dataset using support symmetry, and the corresponding gait can be generated online based on different terrain types. Indeed, this method can generate a variety of non-flat terrain gaits, but due to the different mechanic characteristics of biped robots, the general significance of the dataset cannot be guaranteed. The biped gait data of non-flat terrain is still especially scarce presently, so the applicability of learning-based gait planning is limited. A DC-based gait planning method proposed by Winkler et al. [9] utilized phase-based parameterization of the contact force and the motion of the swing foot to plan gaits for quadrupeds, bipeds and monopeds for various terrains with the robot ANYmal. However, as the dynamic model was still reduced, and the contact model between the supporting foot and terrain is also single point, the planner cannot account for enough motion details for a reasonable gait of complicated motions and more complex robots, such as humanoid robots. Based on the direct collocation, Michael Posa et al. [8] proposed the Constrained Direct Collocation (DIRCON), which can perform gait planning on the implicit constrained state space manifold caused by the derivatives of kinematic constraints. Due to the full-order dynamic model, it is introduced as the collocation constraint, a dynamic feasible joint-level trajectory with third-order accuracy that can be obtained, which enables the complex humanoid robot, Altas, to walk at 1 m/s on flat terrain. Because of the engagement of full-order dynamic with the multi-point contact model, and the derivatives of kinematic constraints, DIRCON is a good potential framework for biped gait planning when there is a need of considering more motion details, such as non-flat terrain walking.

During biped robot movement on non-flat terrain, a coupling relationship exists between the swing foot and terrain such that the height of the swing foot has to be greater than the terrain to avoid collision when the foot reaches the position of the terrain. Moreover, some complex constraints which are difficult to express in linear form can be written in complementary form [19,20,21], such as contact trigger conditions and contact force, which is analogous to the coupling relationship between the swing foot and terrain. Inspired by this, we try to use complementary constraints to introduce the relationships between foot motion and terrains into gait planning. The main contributions of this paper can be summarized as follows: first, we investigate the coupling relationship of terrain information and the biped foot, and the slacked complementary constraint which can describe the different relationships and avoid undesired trajectory are proposed for a TO-based gait planner. Second, with the combination of slacked complementary constraints and cost function, we extend DIRCON into a non-flat terrain biped gait planner which can generate a complete gait or gait sequence with a specific pattern in single planning. Third, the structural design of a newly designed biped robot by our laboratory is validated with the proposed planner through three walking simulations. The work in this paper greatly promotes the development of a novel biped robot, and the proposed planner and constraints can be used as a prototype for high-level motion planning in general bipedal control architectures.

This paper is organized as follows. In Section 2, we recover the theoretical foundations of TO and DIRCON based gait planner. Section 3 first explains how to use a virtual knot to handle the state transitions when the swing foot is touching the terrain, then the coupling relationships between bipedal foot motion and terrain are investigated, and the slacked complementary constraints for gait planning are proposed. Simultaneously, we also show how the elaborated constraints can avoid some undesired trajectories, and then we provide the concrete form of the proposed planner. In Section 4, we test the performance of the proposed planner through three different planar walking experiments: flat terrain walking, stairs terrain walking and quincuncial piles walking; meanwhile, the structural design of our robot is also validated. Section 5 summarizes this paper, analyzes the limitations of the planner and illustrates the future work.

## 2. Related Work

### 2.1. Constrained Hybrid System Dynamics

The robot is a hybrid dynamic system whose state vector typically contains a general position vector $q\in R^{n}$ and a general velocity vector $v\in R^{n}$, i.e., $\dot{x}\left(t\right)={\left[\dot{q}\left(t\right),\;\dot{v}\left(t\right)\right]}^{T}$. The system input u is a d dimensions vector derived from the active joints, and the general form of robotic system dynamics takes:(1)$$\dot{q}\left(t\right)=v\left(t\right),\;\dot{v}\left(t\right)=f\left(q\left(t\right),v\left(t\right),u\left(t\right)\right);\;$$

The biped robot is not only a high-dimensional hybrid dynamic system but also has a constrained floating base. Because the robot needs contact with the terrain or the external environment during movement, contact force should be considered in the dynamic model. Then the dynamics can be further written as:(2)$$\dot{q}\left(t\right)=v\left(t\right),\;\dot{\bar{v}}\left(t\right)=\bar{f}\left(q\left(t\right),v\left(t\right),u\left(t\right),\lambda \left(t\right)\right);\;$$
where λ is the contact force. In this paper the contact model is defined as two forces acting on the front and rear plantar edges of the supporting foot, and the dynamic model can be derived from the data file of our newly designed robot. It is worth noting that the corresponding kinematic position constraints of supporting foot $\varphi \left(q\right):R^{n}\to R^{c}$ should act at the same points of the contact during the whole gait.

### 2.2. Trajectory Optimization and Constrained Direct Collocation

TO can find a feasible joint trajectory because it is naturally capable of dealing with constraints, particularly in high-dimensional and constrained state space. The DC method is currently widely used in biped gait planning. Through the Simpson integration property, DC can insert the dynamics as a collocation constraint at the collocation point, which is usually located at the midpoint between two adjacent knots. The cubic Hermite spline and the first-order curve are utilized to interpolate the state trajectory and the input trajectory, respectively. Thus, DC can avoid the numerical integration process in the Shooting Method [22,23] and the generated trajectory has third-order accuracy, which is the key requirement of stable trajectory tracking [8]. The general form of DC is as follows:(3)$$\underset{z}{\min}\;l_{f}\left(X_{N}\right)\;+\;h\overset{N}{\underset{k=1}{\sum }}l\left(x_{k},u_{k}\right)\;s.t.\;\mathrm{for}\;k=1,2,\ldots N-1\;g\left(x_{k},u_{k},x_{k+1},u_{k+1}\right)=0\;m\left(z\right)\leq 0\;$$
where z is the set of decision variables which consists of $x_{k}$, $u_{k}$ and h. k represents the time index of the state trajectory and the input trajectory, i.e., the kth knot. h is the time step of the adjacent knots and m(z) represents some relevant constraints such as joint limits, input limits, etc. g(z) is the collocation constraint, whose value is equal to the difference between the slope of Hermite spline and robot dynamics at the collocation point of $t_{c,k}$ [20], which is shown as follows:(4)$$t_{c,k}=\frac{1}{2}\left(t_{k}\;+\;t_{k+1}\right),h=t_{k+1\;}-\;t_{k},\;u_{t_{c,k}}=\frac{1}{2}\left(u_{t_{k}}\;+\;u_{t_{k+1}}\right),\;\mathrm{for}\;k=1,2,\ldots N-1\;x_{t_{c,k}}=\frac{1}{2}\left(x_{t_{k}}\;+\;x_{t_{k+1}}\right)\;+\;\frac{h}{8}\left({\dot{x}}_{t_{k}}-{\dot{x}}_{t_{k+1}}\right),\;{\dot{x}}_{t_{c,k}}=\frac{3}{2h}\left(x_{t_{k}}\;-\;x_{t_{k+1}}\right)\;+\;\frac{1}{4}\left({\dot{x}}_{t_{k}}+{\dot{x}}_{t_{k+1}}\right),\;g\left(x_{k},u_{k},x_{k+1},u_{k+1}\right)={\dot{x}}_{t_{c,k}}-f\left(x_{t_{k}},u_{t_{c,k}}\right)$$

In order to make the planned trajectories conform to the system dynamic, DC requires the state variables ($q_{i}$,${\dot{q}}_{i}$,$q_{i+1}$,${\dot{q}}_{i+1}$) at the adjacent knots, which satisfy the collocation constraint and the coefficients in the interpolated splines, can be derived from the states of the knots. In other words, the collocation constraint directs the search of optimization to a local optimal solution of the current cost function, which is dynamically feasible.

Posa et al. [9] found that the derivatives of kinematic position constraints will further compress the state space manifold, so DIRCON was proposed to carry out robot motion planning in this constrained manifold. The specific form of DIRCON is shown as follows:(5)$$\underset{z}{\min}\;l_{f}\left(X_{N}\right)\;+\;h\overset{N}{\underset{k=1}{\sum }}l\left(x_{k},u_{k}\right)\;s.t.\;\bar{g}\left(x_{k},u_{k},x_{k+1},u_{k+1},\lambda_{k},\lambda_{k+1},{\bar{\lambda }}_{k},{\bar{\delta }}_{k}\right)\;=\;{\dot{\bar{x}}}_{t_{c,k}}\;-\left[\begin{array}{c} v+J{\left(q_{t_{c,k}}\right)}^{T}{\bar{\delta }}_{k}\\ \bar{f}\left(x,u,{\bar{\lambda }}_{k}\right) \end{array}\;\right]=0,\;\varphi \left(q\right)=0,\;\dot{\varphi }\left(q\right)=\phi \left(q,v\right)=0,\;\ddot{\varphi }\left(q\right)=\psi \left(q,v,\dot{v}\right)=0,\;for\;k=1,2,\ldots N-1\;m\left(z\right)\leq 0.$$
where $\lambda_{k},\;{\bar{\lambda }}_{k}\;\mathrm{and}\;{\bar{\delta }}_{k}$ are the contact forces, force correction and velocity correction, respectively. ${\dot{\bar{x}}}_{t_{c,k}}$ is the slope of the spline at the collocation point, which has the same form with the ${\dot{x}}_{t_{c,k}}$ in Equation (4), and the location of the collocation point is set to the midpoint between adjacent knots as per convention. As a result, DIRCON provides us a general framework for the complicated motion planning of implicit constrained manifold and maintains the advantages of DC. In addition, due to the explicit introduction of contact force λ, some constraints, such as friction cone, can be easily introduced which is especially suitable for gait planning of biped robots.

## 3. Approaches

### 3.1. Virtual Knots and Multi-Modal Gait Planning

The movement of the biped robot can be regarded as a hybrid system which consists of several single dynamic systems in different phases and the discrete state transitions are triggered by guard conditions. A typical case is that robot will transfer from single support phase to double support phase when the swing foot contacts the ground without slip, and the contact force will cause a sudden change of the state as the swing foot will instantly stop. In order to plan a complete gait (left-right-left) or a sequence of multi-modal gaits (combination of gaits with different speeds), the discrete transitions should be explicitly introduced into the planner. When the contact occurs, the gait planning can be divided into two adjacent modes and the initial state of current mode is related to the last state of the previous mode, i.e., the position is maintained and the velocity abruptly changes, which complies with the impact law during state transition. We use a momentum observer [24] to model the transitions; a virtual knot, which is constrained by the momentum observer, is inserted as the first knot in the trajectory of the current mode. Then, the two mode trajectories can be planned independently. Because of the existence of virtual knots, planners can connect different modes and then plan a complete gait or a gait sequence consisting of multi-modal gaits in a single plan.

For brevity, assuming the target gait consists of two modes, left support phase (LSP) and right support phase (RSP), and the robot is switching from LSP to RSP. The state transitions can be estimated using momentum observer in the following way:(6)$$J{\left(q_{t_{k}}\right)}^{T}\mathrm{Impulse}=K_{O}M\left(q_{t_{k}}\right)\left(v_{\mathrm{post}}-v_{\mathrm{pre}}\right)\;q_{\mathrm{post}}=q_{\mathrm{pre}}=q_{t_{k}}\;$$
where Impulse is the Ground Reaction Force (GRF) at the $t_{k}$ moment of the swing foot in contact with the ground, $v_{\mathrm{post}}$ and $v_{\mathrm{pre}}$ are the velocity components in the state vector before and after the contact. $J\left(q_{t_{k}}\right)$ is the Jacobian matrix of the contact point. $M\left(q_{t_{k}}\right)$ and the $K_{O}$ are the inertia matrix and diagonal gain matrix, respectively. We assume that the LSP has $m_{I}$ knots of modI and the RSP takes $m_{i+1}$ knots of ${\mathrm{mode}}_{i+1}$. In order to maintain trajectory continuity between modes, the last knot of ${\mathrm{modI}}_{i}$ should be the first knot of ${\mathrm{mode}}_{i+1}$ so that the whole sequence contains $\left(m_{i+1}+m_{i}\right)-1$ knots. A virtual knot with the form of Equation (7) is inserted between two modes. As a result, the whole sequence has $\left(m_{i+1}+m_{i}\right)$ knots now, and the virtual knot should satisfy the constraints of Equation (6). Figure 1 depicts the trajectory articulation between the two modes for one dimension after the insertion.
(7)$$Virtual\;knot_{i}=\left[\begin{array}{c} q_{vir}\\ v_{vir} \end{array}\right]=\left[\begin{array}{c} q_{pre}\\ v_{post\_i} \end{array}\right]\;$$

The virtual knot establishes the relationship between two modes when contact happens, thus, the planner can plan the independent trajectory for each mode. The following are the main steps of the DIRCON with virtual knots:

Step 1: Determine the mode sequence according to the contact condition and the robot state. Then establish optimization program based on DIRCON as Equation (5). Turn to Step 2 for virtual knots insertion;

Step 2: Check if current mode happens with contact, if not, stay at Step 2 for next step, or add optimization decision variables ${\mathrm{Impulse}}_{i}$ and $v_{\mathrm{post}\_i}$, respectively and turn to Step 3. If current mode is the last, turn to Step 4;

Step 3: Insert the virtual knots as the initial knot of ${\mathrm{mode}}_{i}$ and organize the equation constraints of the ${\mathrm{Impulse}}_{i}$ and the $v_{\mathrm{post}\_i}$ as Equation (6). If ${\mathrm{mode}}_{i}$ is not the last mode, return to Step 2, or the insertion has completed and go to Step 4;

Step 4: Solve the optimization problem, and restore the joint level trajectories and the input trajectories by interpolation using cubic Hermite spline and first-order curve, respectively.

### 3.2. Slacked Complementary Constraint of Terrain Information

#### 3.2.1. Constraints of Support Location Selection

The location of the supporting foot and the obstacle avoidance of the swing foot are two critical keys during biped motion on non-flat terrain and we will illustrate how the proposed constraints introduce them into the planner, respectively. For brevity, a four-step movement of a seven-link XOZ planar biped robot is used as an example. The selection of foot location is depicted in Figure 2, where the robot foot can be considered as a linear foot with length l, the support terrain consists of four rectangles from $a_{0}$ to $d_{1}$ and $x_{i}$ is the center of the supporting foot. The orange and green area are defined as dangerous supporting terrain of length L that the robot is considered to be at risk of falling when the supporting foot is located here. Therefore, the safety requirement of foot location is to guarantee each selection is within the blue area.

Assuming the selection of ${\mathrm{mode}}_{0}$ is already safe, the three subsequent selections must satisfy the linear inequalities as shown in Equation (8) due to the safety requirement.
(8)$$b_{0}+0.5l\leq x_{1}\leq b_{1}-0.5l\;c_{0}+0.5l\leq x_{2}\leq c_{1}-0.5l\;d_{0}+0.5l\leq x_{3}\leq d_{1}-0.5l$$

#### 3.2.2. Slacked Complementary Constraint of Obstacle Avoidance of the Swing Foot

After foot location selection satisfies the safety requirement, we need to design a constraint to keep the swing foot from colliding with the terrain or obstacles. The swing foot obstacle avoidance is more complicated than foot location selection and the coupling relationship can be divided into three situations according to the height of target terrain and current supporting terrain: swing to flat terrain, swing to higher terrain, and swing to lower terrain. As shown in Figure 3, we continue to utilize the seven-link planar robot model as example.

Swing to higher terrain

When the robot swings its foot to a terrain higher than the current supporting terrain, the front and rear edges which reach the position of the terrain must be higher than the terrain height to avoid collision. Three typical trajectories that satisfy the requirement of collision avoidance of higher terrain are shown in Figure 3a. The dashed gray trajectory first raises the foot higher than the target terrain and then moves horizontally and makes a rapid descent at the end. The black trajectory moves horizontally first, then rapidly lifts, and finally, rapidly descends, and the green trajectory behaves more mildly without significant ascent or descent. All the trajectories satisfy the requirement of collision avoidance as shown in Equation (9). Actually, we frequently select the green as a desired trajectory due to the smoothness, safety and balance maintenance of the biped robot.
(9)$$x_{\mathrm{font}\_\mathrm{end}}\geq e_{1}\;z_{\mathrm{font}\_\mathrm{end}}>d_{4}\;x_{\mathrm{back}\_\mathrm{end}}\geq e_{1}\;z_{\mathrm{back}\_\mathrm{end}}>d_{4}\;$$

Directly utilizing the complementary form, indeed, can guarantee the swing foot passes through the obstacle, but it cannot avoid or reduce the occurrence of the black and gray trajectories. Setting a reasonable cost function of joint input to avoid a huge acceleration is a possible method, however, the value of weight can only be determined through ample real machine experiments, and the process has to keep repeating when any parameter or terrain is changed. In order to induce the solver to converge to the green trajectory, we introduce additional slack variables and reform the complementary constraint of the front edge as follows:(10)$$\left(x+0.5l-e_{1}+{\mathrm{slack}}_{0}\right)\left(z-d_{4}{}^{*}\right)\geq 0,\;(e_{1}-e_{0})>{\mathrm{slack}}_{0}\geq c(e_{1}-e_{0})\;$$

As shown in Figure 3a, x represents the center coordinate of the swing foot and ${\mathrm{slack}}_{0}$ is the slack decision variable to scale the distance between the proximal end of the terrain and the swing foot. c is a positive decimal smaller than 1, $e_{1}$ and $e_{0}$ are the coordinates of the supporting terrain and target terrain, respectively, and $d_{4}{}^{*}$ is a slacked safety height slightly larger than $d_{4}$. This elaborate slacked complementary constraint implies that when the front edge of the foot reaches the point of $e_{1}-{\mathrm{slack}}_{0}$, its height will be greater than the slacked safe height $d_{4}{}^{*}$. Therefore, adding a term of the slack variable ${\mathrm{slack}}_{0}$ to the cost function will induce the solver to converge to the desired pattern trajectory, such as a square term of the difference between the midpoint of the adjacent terrains and ${\mathrm{slack}}_{0}$. Then, the green trajectory can be easily obtained. The constraint of rear edge takes the same form with the front edge and the planner will successfully generate a target gait so that the robot steps to higher terrain safely.

b.Swing to lower and flat terrain

The situation changes when the height of target terrain is lower than the supporting terrain. As shown in Figure 3b, considering the requirement of balance maintenance and the purpose of reducing the horizontal speed when robot foot is in contact with the terrain, the green trajectory in Figure 3b is more reasonable than the gray and the black. Although the black may occur, it is definitely not what we want, and the gray trajectory is acceptable when the contact force does not exceed the friction limitation. To eliminate the black-like trajectory, we can also use the slacked complementary constraint in the following form:(11)$$-\left(x-\left(b_{0}+{\mathrm{slack}}_{1}\right)\right)\left(z-d_{5}{}^{*}\right)\geq 0,\;b_{1}-b_{0}-0.5l>{\mathrm{slack}}_{1}>0.5l\;$$

The purpose of the slack variable ${\mathrm{slack}}_{1}$ is same with Equation (10), $b_{0}$ and $b_{1}$ are the terrain parameters shown in Figure 3b. Through the constraint, the solver ensures that the height of the swing foot does not fall below $d_{5}{}^{*}$ when its x coordinate does not exceed $b_{0}+{\mathrm{slack}}_{1}$. Compared with constraint of the higher terrain, the sign of Equation (11) is opposite due to the target terrain being lower than the supporting terrain, and the black trajectory can be directly avoided even without the effort of a cost function. Due to the supporting terrain, the same trick can be implemented as Equation (12), where the edge should be higher than the supporting terrain until approaching the contact point, which is determined by ${\mathrm{slack}}_{1}$. With the assistance of Equation (8), the solver can easily converge to the desired gait.
(12)−(x−(b0+slack2))(z−d5^)≥0,slack2=c∗slack1 

When the height of the target terrain is equal to the supporting terrain, the requirement becomes quite simple; the Z coordinate of the swing foot is greater than the support terrain at all knots, except the first and the last, and the gait should be similar to the flat terrain gait. However, in order to keep the consistency of the narrative, we still give the form of the slacked complementary constraint:(13)$$\left(x-\left(c_{0}+{\mathrm{slack}}_{3}\right)\right)\left(z-d_{6}{}^{*}\right)\geq 0,\;c_{1}-c_{0}-0.5l>{\mathrm{slack}}_{3}>0.5l\;$$

The ${\mathrm{slack}}_{1}$, ${\mathrm{slack}}_{2}$ and ${\mathrm{slack}}_{3}$ can also be introduced in a cost function with the same form of ${\mathrm{slack}}_{0}$ to achieve a desired pattern. One may notice that the Equation (13) can be written in linear form, which is easier than the complementary form. Indeed, a linear constraint that the height of the swing foot is just greater than the terrain is efficient and the complementary form of flat terrain here is just to keep consistent with the above description due to the types of structural terrains. It is worth noting that we only take the single step movement of a 2D planar biped robot as an example to introduce how the slacked complementary constraint describes the relationship between terrain and biped foot. It is easy to expand from 2D single step to 3D multiple steps because the only needed work is repetition. As the terrain information is the only factor influencing gait pattern, we can generate a multi-step sequence from single step planning when terrain is periodic, such as stairs.

### 3.3. Hybrid DIRCON with Slacked Complementary Constraints

Now we present the complete form of the proposed DIRCON-based non-flat terrain gait planner. As an example, the robot would take three modes to step over two identical stairs and the robot motion is similar to Figure 2 and Figure 3a. Three modes and two contacts exist in the movement, and the entire trajectory consists of N knots after inserting two virtual knots at the beginning of ${\;\mathrm{mode}}_{1}$ and ${\mathrm{mode}}_{2}$. The planner finally takes the following form:(14)$$\underset{z}{\min}\;l_{f}\left(x_{N}\right)+h_{i}\overset{N}{\underset{k=1}{\sum }}l\left(x_{k},u_{k}\right)+K_{\mathrm{high}}{\left({\mathrm{slack}}_{0}-c\left(e_{1}-e_{0}\right)\right)}^{2}+K\overset{N}{\underset{k=1}{\sum }}L\left(x_{k},x_{k-1},u_{k},u_{k-1}\right)\;s.t.\;\bar{g}\left(x_{k},u_{k},x_{k+1},u_{k+1},\lambda_{k},\lambda_{k+1},{\bar{\lambda }}_{k},{\bar{\delta }}_{k}\right)\;=\;{\dot{\bar{x}}}_{t_{c,k}}\;-\left[\begin{array}{c} v+J\left(q_{t_{c,k}}\right){\bar{\delta }}_{k}\\ \bar{f}\left(x_{k},u_{k},{\bar{\lambda }}_{k}\right) \end{array}\;\right]=0,\;\mathrm{for}\;m\;\mathrm{in}\;\left\{{\mathrm{mode}}_{0},{\;\mathrm{mode}}_{1},{\;\mathrm{mode}}_{2}\right\}\;\varphi_{m}\left(q_{k}\right)=0,\;\dot{\varphi_{m}}\left(q\right)=\phi \left(q_{k},v_{k}\right)=0,\;\ddot{\varphi_{m}}\left(q\right)=\psi \left(q_{k},v_{k},u_{k},{\bar{\lambda }}_{k}\right)=0,\;\mathrm{foot}\;\mathrm{selection}\;{\mathrm{constraint}}_{m}=\{\begin{array}{c} X_{k}-b_{0}^{+}\geq 0\\ -\left(X_{k}-b_{1}^{-}\right)\geq 0 \end{array},\;\mathrm{swing}\;\mathrm{foot}\;{\mathrm{constraint}}_{m}=\{\begin{array}{c} \left({\bar{X}}_{k}-\left(e_{1}-{\mathrm{slack}}_{0}\right)\right)\left({\bar{Z}}_{k}-d_{4}{}^{*}\right)\geq 0\\ \left(e_{1}-e_{0})>{\mathrm{slack}}_{0}>c(e_{1}-e_{0}\right)\\ c=0.5 \end{array},\;\;\mathrm{for}\;\ldots \;\mathrm{in}\;\left\{10irtual,10\mathrm{irtual}\right\}\;{\mathrm{Virtual}\;\mathrm{knots}\;\mathrm{constraints}}_{i}=\{\begin{array}{c} q_{\mathrm{vir}\_i}=q_{\mathrm{pre}\_i}\\ v_{vir\_i}=v_{\mathrm{pre}\_i}+M{\left(q_{\mathrm{pre}\_i}\right)}^{-1}K_{O}^{-1}J{\left(q_{\mathrm{pre}\_i}\right)}^{T}{\mathrm{Impulse}}_{i} \end{array}\;\mathrm{for}\;k=1,2,\ldots N-1\;m\left(x_{k},\ldots \right)\leq 0\;\mathrm{Contact}\;\mathrm{friction}\;\mathrm{constraints}\;\mathrm{Periodic}\;\mathrm{constraints}\;\mathrm{Duration}\;\mathrm{constraint}\;\mathrm{Other}\;\mathrm{constraints}$$
where z is the set of all decision variables, $x_{k}{\;\mathrm{and}\;u}_{k}$ are the robot state and joint input at the kth knot. $h_{i}$ are the decision variables of all the time steps between the two adjacent knots. $K_{\mathrm{high}}$ is the weight of trajectory induction term. X, Z and ${\bar{X}}_{k},\;{\bar{Z}}_{k}$ are the slack variables of the supporting foot coordinate and swing foot coordinate at the kth knot in ${\mathrm{mode}}_{I}$, respectively. $q_{\mathrm{vir}\_i}$ and $v_{\mathrm{pre}\_i}$ are the state of the virtual knot at ith contact. $\left[q_{{\mathrm{pre}}_{i}},{\;v}_{{\mathrm{pre}}_{i}}\right]$ is the original knot before the contact and ${\mathrm{Impulse}}_{i}$ is the GRF. In terms of constraints, $\bar{g}\left(\cdot \right)$ is the full order dynamic collocation constraint of DIRCON with multi-point contact model, $\varphi_{m}\left(q\right)$, $\dot{\varphi_{m}}\left(q\right)$, $\ddot{\varphi_{m}}\left(q\right)$ are the kinematic position constraints and their derivatives, respectively, foot selection constraint and floating foot constraint as described in Section 3.2 and the parameters, such as $d_{4}{}^{*},$ should be tuned according to the terrain height of different modes. Virtual knots constraints are used for the inserted knots at the beginning of ${\mathrm{mode}}_{1}$ and ${\mathrm{mode}}_{2}$ as illustrated in Section 3.1. m(z) is the retainable constraint representing joint limits, velocity limits and input limits on the whole trajectory. Contact friction constraints and Periodic constraints are the friction cone constraint of multi points contact model and the gait periodic constraint, respectively. The duration constraint maintains that the total duration of ${\mathrm{mode}}_{0}$ and ${\mathrm{mode}}_{2}$ is equal to ${\mathrm{mode}}_{1}$, and the total duration is also constrained to 1.0 s. Other constraints include the constraints of gait pattern such as the max height of the swing foot, gait speed, upward increment of COM, etc. Due to the cost function, the first two terms are the final state cost and the process state cost during the movement. The third term is the swing foot trajectory induction term and the last term is the trajectory smoothing cost, which is the squares of the difference of robot state and input decision variables in adjacent knots. Due to the contact, we have not predetermined the exact time of contact, and instead we specified a contact sequence. During the complete gait of LSP-RSP-LSP, the contacts occur at the final knot of ${\mathrm{mode}}_{0}$ and ${\mathrm{mode}}_{1}$, respectively. The exact time of the contact are determined by the planner according to the duration constraint and gait speed constraint. It is worth noting that the number of slacked complementary constraints of the floating foot is twice the number of knots due to the two points contact model, and the number of foot location constraints is equal to the number of knots, because we set the constraint to the mid-point of the supporting foot which is not supposed to move. As the constraints should always be satisfied, all the slacked constraints should act on each knot. After solving the optimization, the joint state and input trajectory can be directly obtained.

## 4. Experiments

In this section, the proposed planner as shown in Equation (14) is verified through three planar walking simulations: flat terrain walking, stairs terrain walking and quincuncial piles walking.

### 4.1. Robot Model and Details of Implementation

The robot model used for the simulation is a newly designed biped robot by our laboratory which consists of five parts: base, hip, thigh, shank and foot. Each leg has five active joints, three for the hip, one for the knee and one for the ankle. The connector of joint is made of aluminum alloy, and the main body is made of high-strength carbon fiber. As shown in Figure 4, the robot stands 1.2 m tall and weighs 13.5 kg. The design weight of the base is 3.0 kg plus 2.7 kg for a single hip, which consists of three identical active joints. The thigh weight is 1.1 kg, 1.3 kg for the shank and 0.15 kg for the foot. The robot state space of the robot is 24-dimensional, namely 12-dimensional general position and 12-dimensional general velocity. More technical details of the biped robot will be explained in another paper.

Due to the details of implementation, our hardware used to solve the optimization is “Intel Core i7 9750H” and the software environment is Ubuntu 18.04. C++ and CMake are used as implementation language and compilation tool, respectively. Coppeliasim 4.2.0 [25] is used as the simulation platform and the dynamic engine “Newton” implements the physics. We utilize a robot toolbox “Drake” [26] from MIT to implement the planner that the plant parser is used to establish the dynamic model and the framework of mathematic program is engaged for building the planner. An efficient nonlinear program solver “SNOPT” is used to solve the optimization. Normalized random trajectories, which are modulated according to the parameters of the robot structural design, are introduced as the initial guess. As for other priors, the max joint torques should not exceed 10 Nm, the velocities are also smaller than 6.28 rad/s and the GRF is initialized to the gravity of the robot. After solving, the joint state trajectories and input trajectories are restored and a sample of 100 Hz is conducted to acquire the target gait. A control program which sends the gait and triggers the simulation step-by-step is completed through the Remote API of CoppeliaSim, the robot model files are parsed and the robot is installed into the simulation by a plugin. The control mode of each active joint is set to PD position control and the friction coefficient of terrain is set to 0.6. The time step of the dynamic engine is also set to 10 ms, which matches the frequency of the sample of trajectories. It is worth noting that the dynamic model in the planner is planar that three floating coordinates exist, namely ‘X’, ‘Z’ and ‘Pitch’, and correspondingly, the simulation is also 2D. In order to realize sagittal plane simulation, some constraints are implemented to shun the motion of the coronal plane. A small mass link (0.005 kg) is fixed to the world frame and another two small mass links (0.01 kg), as well as the baselink of the robot, are attached by two prismatic joints and one continuous revolute joint, respectively, and the cascade is shown in the frames of walking. The three joints realize the floating degrees of freedom of sagittal plane. The mass of three links are the experienced values which cannot be zero due to the unknown behavior of physics engine of zero-mass link. Finally, the gait is well tracked in simulation and enables the robot to execute planar walking on different terrains.

### 4.2. Flat Terrain Walking

The first experiment is that the robot walks on flat terrain along the X direction at a speed of 0.5 m/s. At this time, the planner disables the foot location selection constraint and the swing foot constraint. The target gait is designed as a three mode sequence (LSP-RSP-LSP) which consists of 38 knots and 2 virtual knots. The first mode has 10 knots, and 20 knots are in the second. The total duration is constrained to less than 1.0 s. The second LSP also has 10 knots due to the symmetry. Furthermore, a symmetry constraint of the supporting foot location is inserted to ensure that the left support phase and right support phase take the same step length. The quality of the prior trajectory is one of the major factors influencing the performance of planner. We use a normalized random prior, and the SNOPT solving time ranges from 120 s to 3 min. Figure 5 depicts the joint input trajectory generated by the planner, while the target position trajectory is shown in Figure 6 as blue. The maximum joint input is 18.5 Nm and the joint position trajectory is relatively smooth, which can be tracked easily. We use simple PD position control to track the planned trajectory and the performance is shown in Figure 6 as red. Because the collocation constraint in the planner employs a full-order dynamic model, even PD control can track the target trajectory well. The simulation shows that the planned trajectory can keep the robot walking smoothly for all of the simulation time. Figure 7 gives five key gait frames of walking on flat terrain which consists of right foot lifting, stepping forward, right foot contacting the ground, left foot lifting and stepping forward, and these motions form a complete periodic gait. The full simulation can be found in the first section of the Appendix A.

### 4.3. Stepping Upstairs

Walking on stairs is the second experiment. Assume we have obtained the information of stair terrain through some other technical methods that the width and height are 3 cm and 22 cm, respectively. In contrast to walking on flat ground, the planner enables the supporting foot location constraints and swing foot constraints. We also modified the constraint on the height of the robot COM, as well as kinematic position constraints. That is, the height of the robot COM rises 6 cm in Z direction for each step and the supporting foot raises 3 cm for each contact, while the gait speed in X direction remains 0.5 m/s. Other constraints are consistent with the experiment on flat ground. The average time of solving the entire optimization is about 5.5 min. Figure 8 and Figure 9 show the final planned joint position trajectory and input trajectory. The peak value of joint input is 34.7 Nm at this time, which is greater than when walking on flat terrain walking but still less than the rated output torque of the motor (50.0 Nm). Furthermore, although we have tried to adjust the weight of the smoothing term in the cost function, the joint position trajectory cannot behave as smoothly as the gait on flat terrain. The robot continues to track the target trajectory using PD position control. Figure 10 depicts the five key frames of the gait which have the same stepping sequence as Figure 7. According to the simulation, the proposed planner can provide a feasible stair terrain gait which allows the robot to step up the stairs stably. The entire experiment can be found in the second section of the Appendix A.

### 4.4. Quincuncial Piles Walking

The quincuncial piles gait planning is more complicated as the constraints become much more strict, due to managing the narrow supporting area, and ensuring the swing foot does not collide with the terrain at same time. This experiment validates not only the effectiveness of the foot location selection constraint and swing foot constraint in a strict manner, but also the ability to avoid non-ideal trajectory. The distance between two piles is 11 cm, and each pile is 25 cm in length and 14 cm in width. At this point, the planner must adjust the corresponding parameters of the foot location selection constraint and the swing foot constraint based on the terrain information. In addition, we release the gait speed requirement, so that the forward speed in the X direction is just constrained to greater than or equal to 0.5 m/s. After around 5 min of solving, the solver successfully found a trajectory with a speed of 0.59 m/s. Figure 11 depicts the planned joint input trajectory and the peak torque of active joint is around 30.0 Nm, which is smaller than the rated torque; the joint position trajectory in Figure 12 is relatively smooth. The five key frames of quincuncial piles walking are shown in Figure 13, and the gait is similar to the flat terrain walking as the complementary constraint is equivalent to the simple height constraint of the swinging foot on flat terrain as demonstrated in Section 3.3. The entire experimental video is available in the third section of the Appendix A.

## 5. Discussion and Conclusions

As evidenced by the three dynamic simulation experiments in Section 4, the proposed DIRCON based non-flat terrain planner with the virtual knot and the slacked complementary constraint is able to plan a dynamic feasible gait for different non-flat terrains. Even tracked by a simple PD position control, our newly designed biped robot can walk stably in the dynamic simulation environment. In terms of practical application, the planner takes a little longer time to solve the optimization because of the nonlinear dynamic model and the nonlinear slacked complementary constraint, especially the swing foot constraint. The higher non-linearity of the planner leads to a more complex program. In order to ensure the real-time performance of the upper-level planning on a real robot, the terrain information must be fully considered, and plan for sufficient gaits off-line to establish a gait library. Then, the robot can acquire different reference gaits according to the terrain information returned by the detection equipment for gait tracking or a priori to online gait generation. Next, we will combine our real biped robot to simplify the collocation constraint and the swing foot constraint, so as to try to improve the performance of the planner to an on-line level.

## Figures and Tables

**Figure 1 biomimetics-07-00203-f001:**
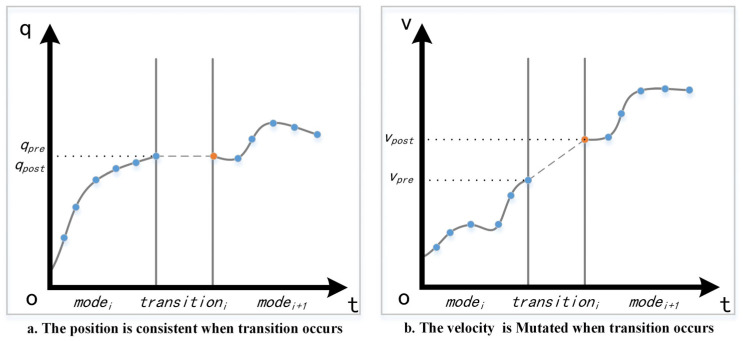
The virtual knot insertion between two adjacent modes where robot state transition occurs when the swing foot contacts the ground. The blue points are the original knot and the orange points are the inserted virtual knot.

**Figure 2 biomimetics-07-00203-f002:**
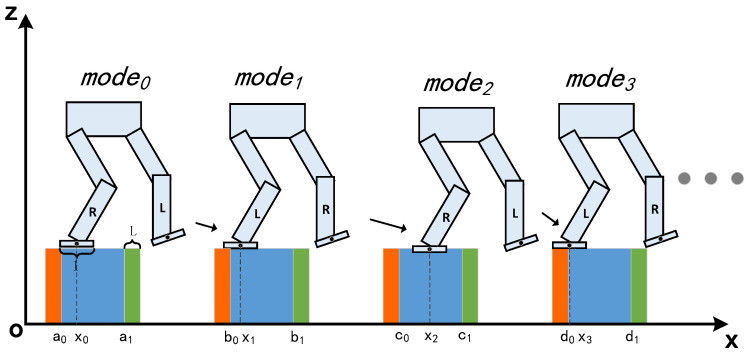
The four selections of supporting foot location where mode0 and mode2 are the desired selections. $x_{0},x_{1},x_{2}{\;\mathrm{and}\;x}_{3}$ are the are the center of supporting foot and $\left[a_{0},a_{1}\right]$, $\left[b_{0},b_{1}\right]$, $\left[c_{0},c_{1}\right]$ and $\left[d_{0},d_{1}\right]$ are the safe areas. If the supporting foot is located in green or orange area, robot will be at the risk of falling.

**Figure 3 biomimetics-07-00203-f003:**
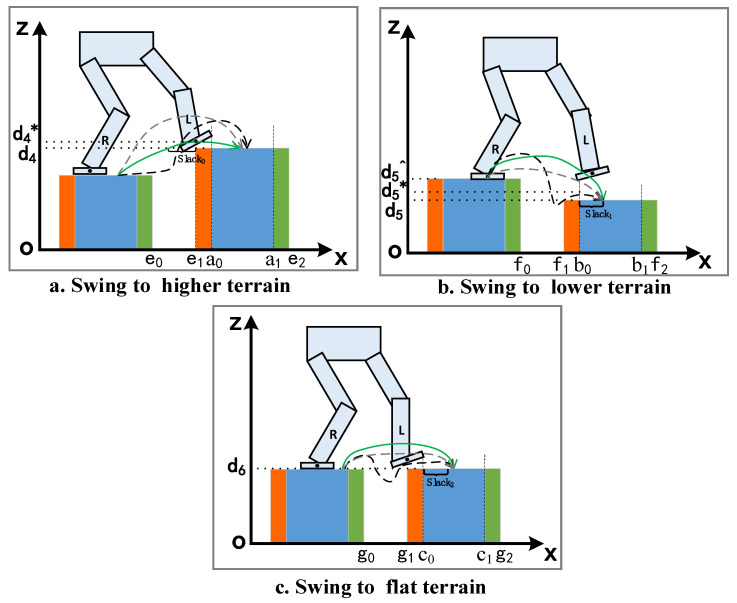
The swing foot movements of different terrain. $d_{4},\;d_{5}{\;\mathrm{and}\;d}_{6}$ are the height of terrain and ${\;d}_{4}^{*},\;d_{5}^{*}\;\mathrm{and}\;d_{6}^{*}$ are the corresponding safe height. d5^ is the height of supporting terrain. The green trajectories are the ideal movement and the gray and the black are the movements which should be avoided.

**Figure 4 biomimetics-07-00203-f004:**
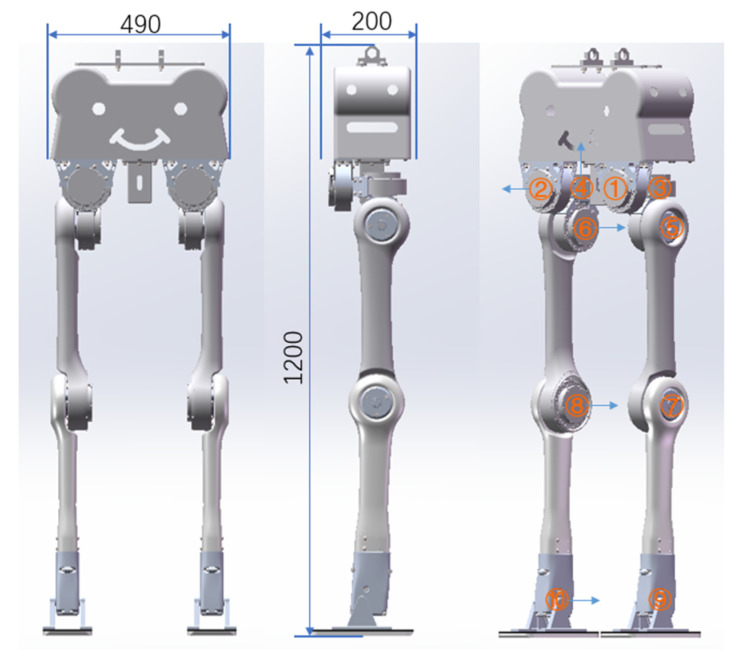
The biped robot model used in this paper. The size of the robot is 490 mm × 200 mm × 1200 mm. The active joints of Roll joint, yaw joint, Pitch joint, knee joint and ankle joint of two legs are numbered from 1 to 10.

**Figure 5 biomimetics-07-00203-f005:**
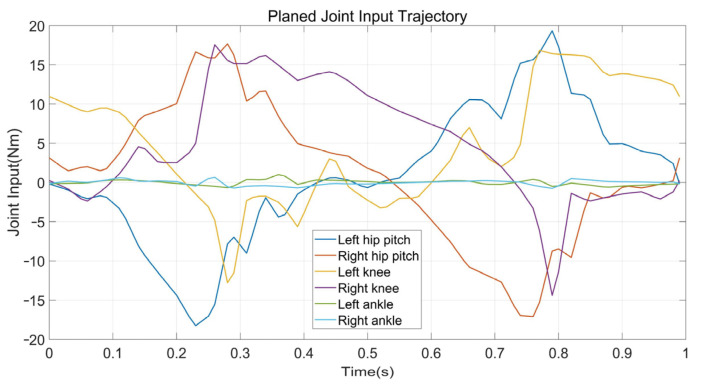
The planned input trajectory of planar walking on flat terrain.

**Figure 6 biomimetics-07-00203-f006:**
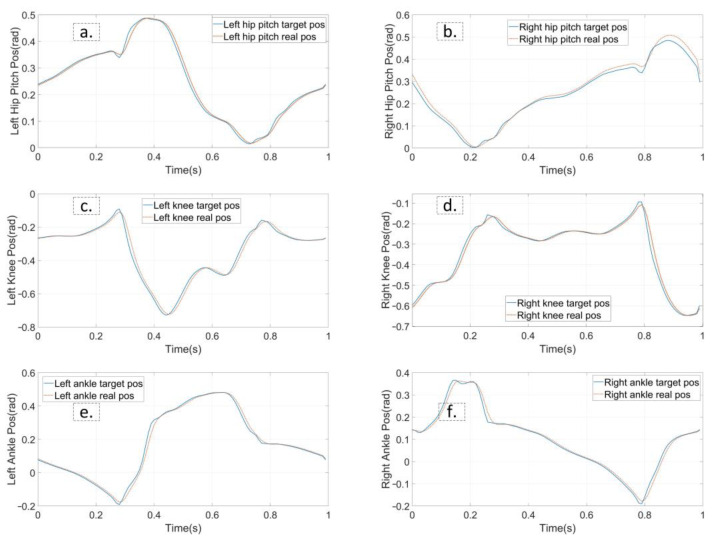
The planned joint position trajectory and the feedback trajectory from simulation. The tracking performance of flat walking simulation is shown from (**a**–**f**). The target position trajectories is represented in blue and real trajectories in red.

**Figure 7 biomimetics-07-00203-f007:**
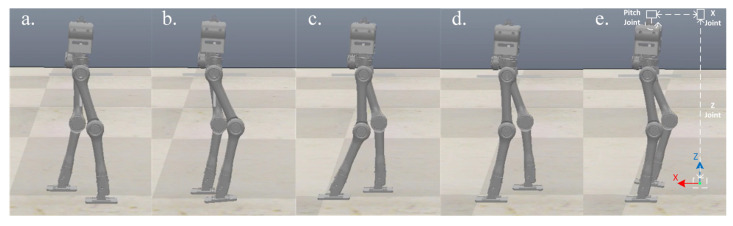
Five key frames of flat terrain walking in CoppeliaSim and a cascade of three small mass links for realizing floating coordinates of sagittal plane. The motion sequence of single complete gait includes right foot lifting, stepping forward, right foot contacting the ground, left foot lifting and stepping forward which is recorded from (**a**–**e**).

**Figure 8 biomimetics-07-00203-f008:**
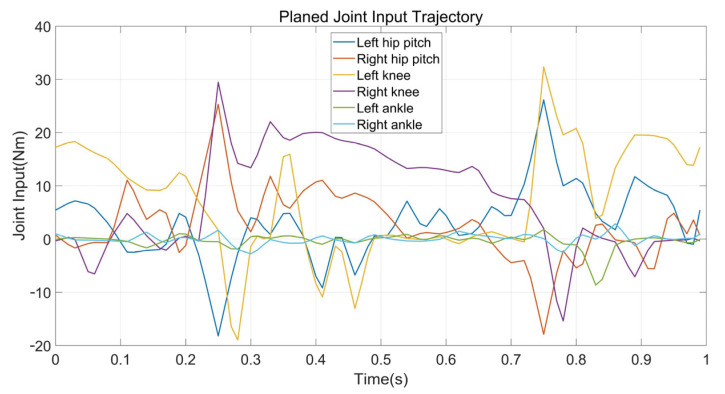
The planned input trajectories of planar walking on stairs.

**Figure 9 biomimetics-07-00203-f009:**
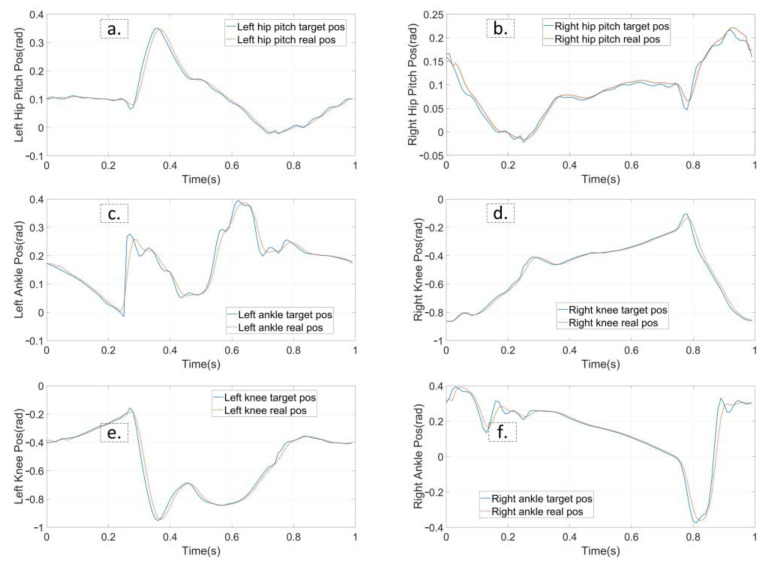
The planned joint position trajectories and the feedback trajectories from simulation of stairs. The tracking performance of each active joint of stepping stairs is shown from (**a**–**f**). The target position trajectories is represented in blue and real trajectories in red.

**Figure 10 biomimetics-07-00203-f010:**
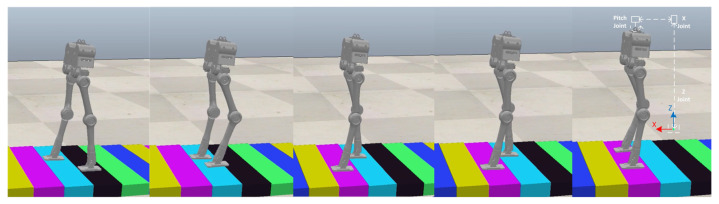
Five key frames of stairs terrain walking in CoppeliaSim.

**Figure 11 biomimetics-07-00203-f011:**
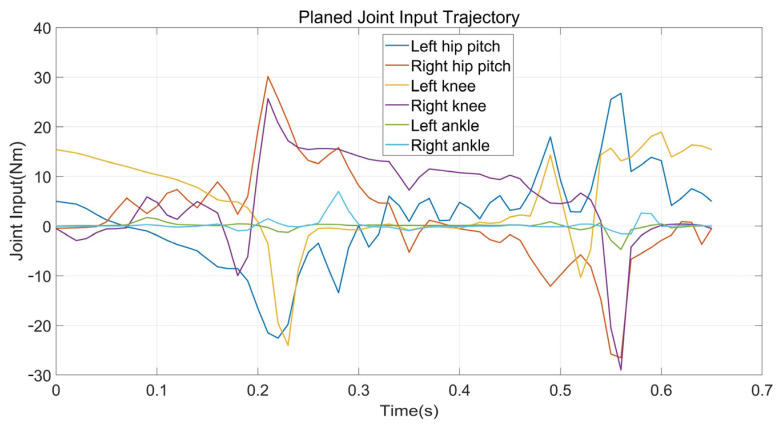
The planned input trajectories of planar walking on quincuncial piles.

**Figure 12 biomimetics-07-00203-f012:**
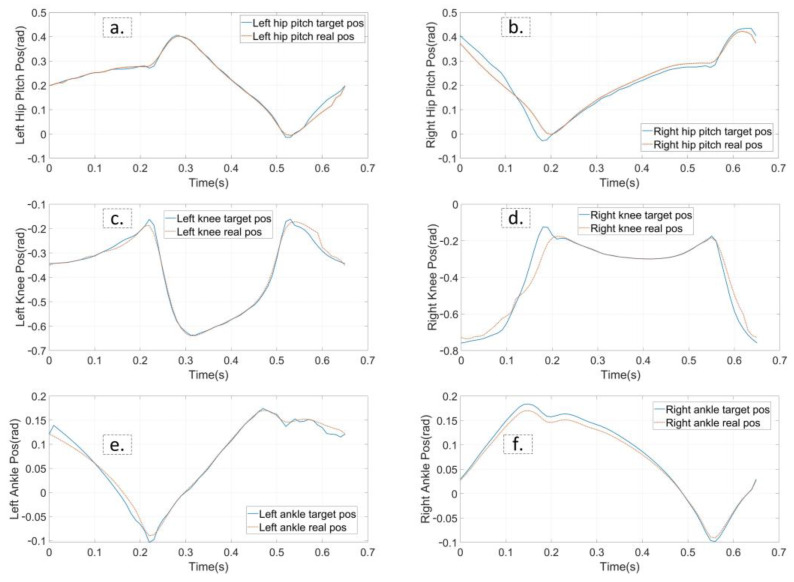
The planned joint position trajectories and the feedback trajectories from simulation of quincuncial piles. The tracking performance of moat walking simulation of each active joint is shown from (**a**–**f**). The target position trajectories is represented in blue and real trajectories in red.

**Figure 13 biomimetics-07-00203-f013:**
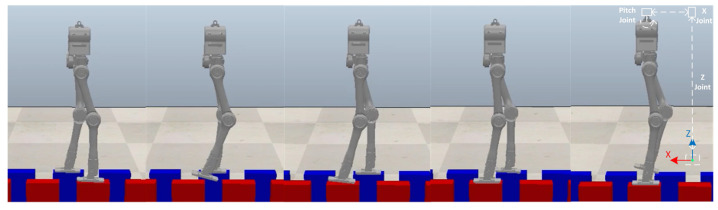
Five key frames of quincuncial piles walking in CoppeliaSim.

## Data Availability

Not applicable.

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
