# Peer review of "A Non-Flat Terrain Biped Gait Planner Based on DIRCON"

_biomimetics, 2022, doi:10.3390/biomimetics7040203_

Round 1
Reviewer 1 Report
This paper presents a direct collocation-based trajectory optimization algorithm for the biped walking problem on a non-flat terrain. The authors used the existing DIRCON tool to perform the trajectory optimization work and used the Coppeliasim to verify the robot motion. Although the authors presented the work in a clear and organized manner with a detailed explanation on the direct collocation, the weakest points of this paper are its unclearness of novelty (please clarify) and its lack of hardware experiments. I have several major comments for the authors’ consideration.
1. Direct collocation is a mature technique for trajectory optimization on legged locomotion. As the author mentioned, Posa et al developed the DIRCON tool to automate this calculation. So I wonder about the novelty and contribution of this work. Please highlight the novelty and contributions of this paper in the introduction.
2. The computation related details of the planning process are missing. For instance, what is the computational environment for the optimization process? How did you choose the initial guess? Could the author provide more details on what actual work was done on Drake, DIRCON, and Coppeliasim respectively.
3. What is the dynamics simulation environment? For instance, which physics engine does Coppeliasim use in the simulation and what method does the physics engine use for the dynamics simulation. Note that these are the important details to evaluate the feasibility of the planned trajectory when the authors are not able to perform a hardware experiment.
4. The authors implemented all the constraints in the complementary form. I don’t know the intention behind this but in my opinion, many of the linear constraints should not be written in a complementary form because the original linear form is easier to be solved for the optimization solver. Posa and other researchers using the complementary form is because there is no linear form to model the contact events. It is unnecessary to model everything in a complementary form.
Some minor issues regarding the technical aspects of the manuscript:
5. Equations 8 and 9 are not equivalent in mathematics. For instance, when x1 = b0, x2 = c0, and x3 = (d0 + d1) / 2, it satisfies Eq. 9 but not Eq. 8.
6. It is not clear if the authors eventually used a 3D model for the trajectory optimization. Was the simulation also in 3D or were there some constraints on the robot motion?
7. Is the contact event predetermined or what information was known prior to the optimization. I did not see time as a decision variable in the optimization formulation. Does this mean that the contact event time is predetermined?
Reviewer 2 Report
This paper presents how to use the slacked complementary constraints to introduce biped robot foothold selection and the relationship between swing foot and terrain into biped gait planning. A DIRCON based biped gait planner for non-flat terrain walking is proposed and the generated gaits can enable the robot to walk on flat terrain, stairs, and quincuncial piles.
the whole work is novel and profitable for full-dynamic biped gait planner of non- flat terrain, and provide an easy-to-use and promising method in this area. The background, approach and verification are also complete. However, some details need to be added and explained clearly, and the paper needs proper English editing.
1. In the background, please add more recent literatures about trajectory optimization based gait planning.
1. How many slacked constraints of terrain information are needed for single planning? Are all of the slacked complementary constraints act on each knot?
2. In the experiment, which dynamic engine is used in the simulation? The hardware of solving the planner should be presented. Please add to the paper.
3. The English needs to be refined.
Round 2
Reviewer 1 Report
This reviewer thanks the authors for addressing my comments. I don't have further questions. If possible, could the authors visualize the constraint links added to the simulation so that the readers understand the simulation results easier?
One more thing is that when I read the revisions, I still found several language issues, such as "From another hand" etc. Please proofread again including the revised part before publication.
